# HBx 128–133 Deletion Affecting HBV Mother-to-Child Transmission Weakens HBV Replication via Reducing HBx Level and CP/ENII Transcriptional Activity

**DOI:** 10.3390/v14091887

**Published:** 2022-08-26

**Authors:** Yarong Song, Ying Lu, Yi Li, Minmin Liu, Hui Zhuang, Jie Li, Jie Wang

**Affiliations:** 1Department of Microbiology & Infectious Disease Center, School of Basic Medical Sciences, Peking University Health Science Center, Beijing 100191, China; 2NHC Key Laboratory of Medical Immunology/Immunology Research Platform of Peking University, Beijing 100191, China

**Keywords:** hepatitis B virus, mother-to-child transmission, X region mutation, HBV replication, hepatocyte nuclear factor 4α

## Abstract

Some infants born to hepatitis B surface antigen (HBsAg)-positive mothers, especially born to hepatitis B e antigen (HBeAg)-positive mothers, can still be infected with hepatitis B virus (HBV) through mother-to-child transmission (MTCT) of HBV and develop chronic HBV infection. At present, the virological factors affecting HBV MTCT are still unclear. In this study, we found that the mutation rates of amino acids in the HBV X region were high, and there were obvious differences between the immunoprophylaxis success group and the immunoprophylaxis failure group of HBeAg-positive mothers. Specifically, the mutation rate of HBx 128–133 deletion (x128–133del) or corresponding nucleotide 1755–1772 deletion (nt1755–1772del) in the immunoprophylaxis success group was significantly higher than that in the immunoprophylaxis failure group. Furthermore, we found that x128–133del could weaken HBV replication by reducing the level of the HBx protein due to the increased proteasome-dependent degradation of HBx protein, and the transcriptional activity of HBV core promoter (CP)/enhancer II (ENII) due to the attenuated binding capacity of hepatocyte nuclear factor 4α (HNF4α) to HBV CP/ENII. This study suggests that x128–133del may contribute to immunoprophylaxis success, which may be helpful in clarifying the virological mechanism affecting HBV MTCT and formulating an optimal immunization strategy for children born to HBeAg-positive mothers.

## 1. Introduction

Chronic hepatitis B (CHB) caused by hepatitis B virus (HBV) infection is one of the most prevalent infectious diseases worldwide. The World Health Organization (WHO) estimated that 296 million people were living with chronic HBV infection around the world in 2019, and there are about 1.5 million new infections and an estimated 820,000 deaths due to HBV-related end-stage liver disease each year [1]. A large portion of the global burden of CHB can be attributed to the mother-to-child transmission (MTCT) of HBV, especially in areas with a high prevalence of HBV. Although the combined immunization with hepatitis B vaccine (HepB) and hepatitis B immunoglobulin (HBIG) can significantly reduce the incidence of HBV MTCT and lead to immunological protection in the vast majority of infants born to hepatitis B surface antigen (HBsAg)-positive mothers, 8–30% of infants born to mothers with high viral load still experience immunoprophylaxis failure [1,2]. Importantly, the rate of chronicity in newborns infected with HBV is higher than that in other HBV-infected people, and may suffer an impaired quality of life [3]. In 2016, the World Health Assembly (WHA) proposed the goal of eliminating viral hepatitis as a public health threat by 2030 [4]. Considering that HBV MTCT leads to a 1/3 of HBV new infection in the world [5], a high priority should be taken to prevent HBV MTCT in the drive toward the elimination of hepatitis B.

Previous studies on the virological factors related to HBV MTCT have mainly focused on the mutations in the HBV “a” determinant region, which might affect the recognition and affinity of anti-HBs to HBsAg, thereby resulting in immune escape contributing to immunoprophylaxis failure [6,7,8,9]. However, many studies do not support the role of the “a” determinant region mutation in the immunoprophylaxis failure of HBV MTCT [10,11,12]. Another possible virological factor of HBV MTCT is the maternal hepatitis B e antigen (HBeAg) status reflecting the HBV viral load, where it has been reported that positive maternal HBeAg is related to the immunoprophylaxis failure of HBV MTCT and could be affected by mutations of the HBV precore (preC) region and basic core promoter (BCP) region [13,14,15,16]. G1896A and A1762T/G1764A are the hot-spot mutations of the preC/BCP region, which have been reported to be able to reduce the risk of HBV MTCT [17,18,19,20,21]. However, the hot-spot mutations of the preC/BCP region in mothers with immunoprophylaxis success were similar to that in the mothers with immunoprophylaxis failure when maternal HBeAg status was strictly matched between the two groups [7].

With the development of genome sequencing technology, the role of HBV X region mutations in the immunoprophylaxis effect of HBV MTCT has been noted. Several studies have reported that the mutation rates of HBV X region in the mothers with immunoprophylaxis failure are different to those in the mothers with immunoprophylaxis success, indicating that HBV X region mutations may affect the immunoprophylaxis effect of HBV MTCT [12,22,23]. The HBV X region encodes a key regulator called HBV X protein (HBx), which consists of two domains, one is the N-terminal domain, which consists of amino acids (aa) 1–50 and can inhibit HBx activities, and the other one is the C-terminal domain, which consists of aa 51–154 and is essential for the transactivation functions of HBx [24]. HBx can promote HBV replication depending on its C-terminal domain, and is required to initiate and maintain HBV infection [25,26,27,28]. In addition, the HBx coding region overlaps with the HBV core promoter (CP) and enhancer II (ENII), thus mutations of the HBV X region may affect HBV replication not only by affecting the HBx protein itself, but also by affecting the transcriptional activity of HBV CP/ENII.

In this study, HBV X region mutations associated with the immunoprophylaxis effect of HBV MTCT were explored on the basis of a large prospective cohort of mother–infant pairs with positive maternal HBsAg, and the functional analysis of the HBx128–33 deletion (x128–133del) or the corresponding nucleotide 1755–1772 deletion (nt1755–1772del) was further performed to explore the effect and mechanism of HBV X region mutations on HBV replication to clarify the virological factors affecting HBV MTCT.

## 2. Materials and Methods

### 2.1. Subjects

Pregnant women recruited from 2009 to 2011 in Jiangsu and Henan Provinces and from 2012 to 2014 in Jilin Province were routinely tested for HBsAg during the first prenatal visit by 12 weeks of gestation. Written informed consent was signed by all subjects. A total of 1694 HBsAg-positive pregnant women without antiviral treatment were enrolled. All infants received three doses of recombinant yeast-derived HepB (10 μg/0.5 mL or 20 μg/1.0 mL) at birth (within 24 h), 1, and 6 months, combined with one dose of HBIG (100 or 200 IU) within 24 h of birth. A total of 1377 infants returned for post-vaccination serologic testing (PVST) at 7 months old. The PVST results for infants at 7 months old were interpreted as: (1) immunoprophylaxis failure: HBsAg was positive and anti-HBs < 10 mIU/mL; (2) immunoprophylaxis success: HBsAg was negative and anti-HBs ≥ 10 mIU/mL. A total of 29 infants had immunoprophylaxis failure, and all were born to HBeAg-positive mothers with high viral loads (>7 log_10_ IU/mL); 22 of them were born to mothers infected with genotype C2 HBV. To reduce the influence from the high heterogeneity of different genotypes, the 22 infants and their paired mothers were investigated. In the immunoprophylaxis success group, 249 mother–infant pairs were eligible for comparison with the 22 mother–infant pairs in the immunoprophylaxis failure group, based on the maternal age, HBsAg titers, HBeAg status, HBV genotype, serum HBV DNA level, ALT level and infantile weight, immunization program, parturition manner, and feeding pattern.

### 2.2. Amplification, Sequencing, and Sequence Analysis

The extraction and amplification of HBV genomes have been described in a previous study [9]. The sequences of the HBV full-length genome were obtained by direct sequencing using seven sequencing primers as described in [9] and were divided into seven coding regions. The details and nucleotide sites of these regions are provided in Appendix A. The HBV X region was amplified by nested PCR and sequenced by direct sequencing. The primers for nested PCR and sequencing are listed in Appendix A. The amino acid sequences were translated on the basis of nucleotide sequences, and mutations were identified by a consensus sequence synthesized by all HBV full-length genomes from the mothers.

### 2.3. Plasmids

The mutant plasmid carrying the x128–133del (pBB4.5-HBV1.2-Del) was constructed by site-directed mutation using pBB4.5-HBV1.2, which contained a wild 1.2-fold length HBV genome of genotype C2 as a template. The HBx expression plasmid (pCDH-HBx-3×flag) was constructed by inserting the HBV X coding region with 3×flag tag sequence into the pCDH vector after digestion with EcoRI and XbaI. The x128–133del mutant HBx expression plasmid (pCDH-HBx-Del-3×flag) was constructed by site-directed mutation using pCDH-HBx-3×flag as the template. The HBV CP/ENII luciferase reporter plasmid (pGL3-CP/ENII) was kindly provided by Dr. Xiang (Peking University, Beijing, China). The nt1755–1772del mutant HBV CP/ENII plasmid (pGL3-CP/ENII-Del) was constructed by site-directed mutation using pGL3-CP/ENII as the template. The hepatocyte nuclear factor 4α (HNF4α) expression plasmid (pCDH-flag-HNF4α) was constructed as previously described [29]. The primers used for plasmid construction are listed in Appendix A.

### 2.4. Antibodies and Reagents

Horse anti-HBs (ab9193), rabbit anti-HBx (ab39716), mouse anti-HNF4α (ab181604), rabbit anti-actin (ab179467), and HRP-linked rabbit anti-horse IgG (ab6921) antibodies were purchased from Abcam (Cambridge, MA, USA). The rabbit anti-preS1 (10R-10460) antibody was purchased from Fitzgerald (Acton, MA, USA). The HRP-linked goat anti-rabbit IgG (BE0101-100) and HRP-linked goat anti-mouse IgG (BE0102-100) antibodies were purchased from Easybio (Beijing, China). The proteasome inhibitor MG132 (S2619) and lysosome inhibitor HCQ (S4430) were purchased from Selleck (Houston, TX, USA).

### 2.5. Cell Cultures and Transfection

Huh7 and HepG2 cells were maintained in Dulbecco’s modified Eagle medium (DMEM) added with 10% fetal bovine serum (FBS) and 1% penicillin/streptomycin (PS) in a 5% CO_2_ incubator at 37 °C. The cells were seeded 18–24 h before transfection, then transfected with plasmids using the Lipofectamine 2000 transfection reagent (Invitrogen, Carlsbad, CA, USA) according to the manufacturer’s protocol. The cells were washed with Dulbecco’s phosphate-buffered saline (PBS) 3 times at 4–6 h after transfection, and then fresh media (DMEM + 2% FBS + 1% PS) was added.

### 2.6. Detections of HBsAg and HBeAg in Supernatants

HBsAg and HBeAg were detected by chemiluminescence immunoassay kits (Autobio diagnostics Co., Zhengzhou, Henan, China) according to the manufacturer’s protocol.

### 2.7. HBV DNA and HBV RNA Quantification

At 72 h post transfection with 3 ug HBV plasmid (pBB4.5-HBV1.2 or pBB4.5-HBV1.2-Del) in Huh7 and HepG2 cells (5 × 10^5^/well in 6-well plate), the supernatants and cells were harvested for HBV DNA and total RNA extraction, respectively. Supernatants were treated with Dnase I (Takara, Tokyo, Japan) for 30 min at 37 °C, and then HBV DNA was extracted using a QIAamp DNA Blood Mini Kit (Qiagen, Hilden, Germany) according to the manufacturer’s protocol. Total RNA was extracted by Trizol Reagent (Thermo Fisher Scientific, Waltham, MA, USA) according to the manufacturer’s protocol and quantified by NanoDrop (Thermo Fisher Scientific, Waltham, MA, USA). Extracted RNA was reverse transcribed to cDNA by the RevertAid First Strand cDNA Synthesis Kit (Thermo Fisher Scientific, Waltham, MA, USA). The HBV DNA and RNA levels were both detected by quantitative real-time polymerase chain reaction (qPCR) with the SYBR Green method. The primers for measuring HBV DNA, HBV 3.5 kb RNA, and HBV total RNA are listed in Appendix A.

### 2.8. Western Blot

The transfected cells were harvested from 6-well plates and lysed with RIPA buffer in the presence of proteinase inhibitors (Roche, Basel, Kanton Basel, Switzerland) on ice for 30 min. The cell lysates were loaded and separated by 12% SDS-PAGE, followed by transferring to a PVDF membrane. Then, the blotted membrane was blocked in 5% skim milk for 1 h and incubated with the primary antibodies at 4 °C overnight. After washing in TBST three times, the membrane was incubated with the secondary antibodies for 1 h at room temperature. Finally, the levels of proteins were detected by a ChemiDoc XRS Imaging System (Bio-rad, Hercules, CA, USA).

### 2.9. Northern Blot

Intracellular total RNA was extracted by the Trizol reagent (Thermo Fisher Scientific, Waltham, MA, USA) according to the manufacturer’s protocol. Northern blot was performed using the DIG Northern Starter Kit (Roche, Basel, Kanton Basel, Switzerland). Briefly, after heat denaturation for 10 min at 65 °C, the extracted RNA was separated in a 1.5% agarose gel containing MOPS and formaldehyde, and then transferred onto a Nylon membrane (Roche, Basel, Kanton Basel, Switzerland). The membrane was hybridized with a Dig-labeled HBV probe, followed by anti-digoxigenin-AP Fab fragments (Roche, Basel, Kanton Basel, Switzerland) incubation for 1 h at room temperature. The probe was generated by PCR with the forward primer: 5′-CTAATCATCTCATGTTCA-3′, reverse primer: 5′-GGACTGCGAATTTTGGCC-3′, and digoxigenin-11-dUTP. The levels of HBV RNA were detected using a ChemiDoc XRS Imaging System (Bio-Rad, Hercules, CA, USA).

### 2.10. Luciferase Reporter Assay

The Huh7 and HepG2 cells (5 × 10^5^/well in 6-well plate) were co-transfected with 3 μg 1.2× HBV plasmid (pBB4.5-HBV1.2 or pBB4.5-HBV1.2-Del) and 0.5 μg nano-luciferase expression plasmid (pCDH-Nluc), which was used as the internal control plasmid. Cell culture supernatants were harvested at 72 h after transfection, and then the luciferase activity was measured using a Nano-Glo Luciferase Assay Kit (Promega, Madison, WI, USA) according to the manufacturer’s protocol.

The Huh7 and HepG2 cells (1 × 10^5^/well in 12-well plate) were co-transfected with 0.5 μg firefly luciferase reporter plasmid (pGL3-CP/ENII or pGL3-CP/ENII-Del), 1 μg pCDH-flag-HNF4α plasmid, and 0.025 μg pRL-TK plasmid carrying renilla luciferase. At 48 h post transfection, the cells were lysed with passive lysate buffer (Promega, Madison, WI, USA), and then the relative luciferase activity was measured using a Dual-Luciferase Assay Kit (Promega, Madison, WI, USA) according to the manufacturer’s protocol.

### 2.11. Chromatin Immunoprecipitation (ChIP)-Polymerase Chain Reaction (PCR)

The ChIP assay was performed using a Simple ChIP Enzymatic Chromatin IP Kit (Cell Signaling Technology, Danvers, MA, USA). In brief, Huh7 and HepG2 cells (3 × 10^6^/well in 10 cm dish) were co-transfected with the 4 μg pGL3-CP/ENII or pGL3-CP/ENII-Del plasmid and 8 μg pCDH-flag-HNF4α plasmid. At 48 h post transfection, cells were crosslinked with 1% formaldehyde at room temperature for 10 min, and then glycine was added to terminate the reaction. After washing with cold PBS, cells were lysed with SDS lysis buffer, and then the lysates were fragmented by sonication to generate DNA fragments and incubated with the mouse anti-HNF4α antibody (Abcam, Cambridge, MA USA) or rabbit IgG (Cell Signaling Technology, Danvers, MA, USA). After immunoprecipitation, the chromatins were purified and then amplified by PCR with the forward primer: 5′-AACGACCGACCTTGAGGC-3′ and reverse primer: 5′-GAACATGAGATGATTAGGCAGAG-3′. The size of the PCR product was about 150 bp.

### 2.12. Statistical Analysis

Statistical analyses were performed by SPSS 24.0 (SPSS Inc., Chicago, IL, USA) software. Normal distributions data were expressed as the mean ± standard deviation and compared by the Student’s *t*-test. Non-normal distributions data were expressed as median (range) and compared by the Mann–Whitney U-test. Categorical variables were expressed as proportions (%, n/n) and analyzed by Chi-square/Fisher’s exact test or the Kruskal–Wallis H-test. Correction for multiple comparison was analyzed by the Benjamini–Hochberg test. *Q* < 0.05 was considered statistically significant. All *p* values were two-tailed, and *p* < 0.05 was considered statistically significant.

## 3. Results

### 3.1. Baseline Characteristics between the Immunoprophylaxis Failure and Success Group

Among the 249 mother–infant pairs in the immunoprophylaxis success group, 22 of them were randomly selected for a comparison with 22 mother–infant pairs in the immunoprophylaxis failure group at the ratio of 1:1 to analyze the full-length HBV genome sequences. As shown in Appendix A, no significant differences in the baseline characteristics were found between 249 mother–infant pairs and the 22 randomly selected mother–infant pairs, suggesting that the randomly selected 22 mothers could represent the 249 mothers in the immunoprophylaxis success group. All mothers were both HBsAg and HBeAg positive in two groups. The baseline characteristics of 22 mothers and their infants in two groups are shown in Table 1. No significant differences in the baseline characteristics were found between the immunoprophylaxis failure and success group.

Furthermore, the mothers in the immunoprophylaxis success group were added to 120 cases to analyze the sequences of the HBV X region. The baseline characteristics of 120 mothers and their infants are shown in Appendix A. In this context, there was also no significant differences in the baseline characteristics between the immunoprophylaxis failure and success group.

### 3.2. X128-133del May Affect HBV MTCT

Amino acid mutations in the HBV coding regions were analyzed on the basis of the full-length HBV genome sequences from mothers in the immunoprophylaxis failure and success groups. The results revealed that all amino acid mutation rates were lower than 10% in either the preC (Figure 1A) or C region (Figure 1B). Although high mutation rates (>10%) were found in the preS1 (Figure 1C), preS2 (Figure 1D), S (Figure 1E), and RT (Figure 1F) regions, there were no obvious differences in the mutation rates between the immunoprophylaxis failure and success groups. In the X region, there were not only high mutation rates (>10%), but also obvious differences in the mutation rates between the immunoprophylaxis failure and success group (Figure 1G), which suggests that HBx mutations might play an important role in the immuprophylaxis effect of HBV MTCT.

Next, HBx mutations were screened by the following standards: (1) HBx mutations only present in the mothers of the immuprophylaxis failure or success group, and mutation rates were higher than 5%; or (2) HBx mutations present in two groups, but the difference in the mutation rates between the two groups was more than twice. As shown in Table 2, xG22S (ntG1437A) and xS42Y (ntC1498A/A1499C) were only present in the immunoprophylaxis failure group; xS31P (ntT1464C), xM103L (ntA1680C), and x128–133del (nt1755–1772del) were only present in the immunoprophylaxis success group; xA23T (ntG1440A) and xG32R (ntG1467C) were mainly present in the immunoprophylaxis failure group; and xD48N (ntG1515A) and xH86R (ntA1630G) were mainly present in the immunoprophylaxis success group. It was worth noting that the mutations were mainly located at the N-terminal domain of the HBx protein in the immunoprophylaxis failure group, whereas the mutations were mainly located at the C-terminal domain of the HBx protein in the immunoprophylaxis success group. Furthermore, the mutation rates of xG22S, xA23T, xG32R, and xS42Y in the immunoprophylaxis failure group was higher than that in the immunoprophylaxis success group, and the mutation rate of x128–133del in the immunoprophylaxis failure group was lower than that in the immunoprophylaxis success group. However, a significant difference in the mutation rate was only found for x128–133del between the immunoprophylaxis failure and success groups (0% vs. 27.50%, *Q =* 0.045) after correction for multiple comparison, suggesting that x128–133del might affect the immunoprophylaxis effect of HBV MTCT. No significant difference in the incidence of hot-spot A1762T/G1764A double mutation was found between the immunoprophylaxis failure and success groups [13.64% (3/22) vs. 6.67% (8/120), *p* = 0.490].

### 3.3. X128–133del Weakens HBV Replication

Since x128–133del was only present in the immunoprophylaxis success group and not in the immunoprophylaxis failure group, it indicated that x128–133del might not be conducive to HBV MTCT by affecting HBV replication. To test this possibility, the HBV replication-competent plasmid carrying the x128–133del mutation (pBB4.5-HBV1.2-Del) was first generated from the corresponding plasmid carrying wild-type 1.2-fold genotype C2 HBV genome (pBB4.5-HBV1.2), and then the pBB4.5-HBV1.2-Del or pBB4.5-HBV1.2 plasmid and nano-luciferase expression plasmid (pCDH-Nluc), which was used as an internal control plasmid, were co-transfected into Huh7 and HepG2 cells to explore the effect of x128–133del on HBV replication.

As shown in Appendix A, the nano-luciferase activity between the wild-type and x128–133del mutant groups was comparable in the Huh7 and HepG2 cells, suggesting that the transfection efficiency was comparable between the two groups. X128–133del could significantly reduce the level of HBsAg in the cell culture supernatants of both the Huh7 and HepG2 cells (Figure 2A). In line with this, x128–133del reduced the levels of large HBsAg (L-HBsAg) and small HBsAg (S-HBsAg) including gp27 and p24 proteins in both the Huh7 and HepG2 cells (Figure 2B). Meanwhile, x128–133del could also significantly reduce the levels of HBeAg and HBV DNA in the cell culture supernatants of both the Huh7 and HepG2 cells (Figure 2C,D). For HBV RNA, RT-qPCR confirmed that x128–133del could significantly reduce the levels of HBV total RNA and HBV 3.5 kb RNA in both the Huh7 and HepG2 cells (Figure 2E,F), which was further confirmed by Northern blot analyses (Figure 2G). These results suggest that x128–133del could weaken HBV replication.

### 3.4. Wild-Type HBx Can Rescue the x128–133del-Weakened HBV Replication 

To further explore whether the weakened HBV replication was mediated by the mutant HBx protein, the wild-type HBx protein was used to try to rescue the weakened HBV replication. The pBB4.5-HBV1.2 or pBB4.5-HBV1.2-Del plasmid, pCDH-HBx-3×flag or pCDH vector control plasmid, and internal control (pCDH-Nluc) plasmid were co-transfected into the Huh7 and HepG2 cells, and then the levels of HBsAg and HBeAg in the culture supernatants of Huh7 and HepG2 cells were detected to analyze the effect of wild-type HBx protein on the x128–133del-weakened HBV replication. The results revealed that the transfection efficiency among the groups was comparable in Huh7 and HepG2 cells (Appendix A). The x128–133del-mediated reductions of HBsAg and HBeAg levels could be partially rescued by the wild-type HBx protein (Figure 3A,B), indicating that the x128–133del mutant HBx protein might contribute to the reduction of HBV replication.

### 3.5. X128-133del Reduces the Level of HBx Protein through Promoting the Proteasome-Dependent Degradation

To further explore the mechanism on the x128–133del-mediated reduction in HBV replication, we first detected the effect of x128–133del on the HBx protein level. The x128–133del mutant HBx expression plasmid (pCDH-HBx-Del-3×flag) was first generated from the wild-type HBx expression plasmid (pCDH-HBx-3×flag), and then the pCDH-HBx-Del-3×flag or pCDH-HBx-3×flag plasmid was transfected into the Huh7 and HepG2 cells to explore the effect of x128–133del on the level of the HBx protein. The results showed that x128–133del could reduce the level of the HBx protein (Figure 4A), but not affect the level of HBx mRNA (Figure 4B), suggesting that x128–133del could reduce the level of HBx protein at the post-transcriptional level.

Since HBx aa82–154 has been reported to mediate the binding of the HBx protein to proteasome subunits and is related to the degradation of the HBx protein [30], the reduction in the HBx protein level might be due to the increased degradation of the x128–133del mutant HBx protein. To determine the influence and pathway of x128–133del on the HBx protein degradation, pCDH-HBx-3×flag or pCDH-HBx-Del-3×flag plasmid was transfected into the Huh7 and HepG2 cells, and then the levels of the HBx protein in the transfected cells were analyzed after the treatment of the proteasome inhibitor MG132 or lysosome inhibitor HCQ. As shown in Figure 4C, the level of the x128–133del mutant HBx protein increased to a level close to that of the wild-type HBx protein under MG132 treatment, but not for HCQ treatment. These results suggest that x128–133del reduced the level of HBx protein by promoting the proteasome-dependent degradation of the HBx protein.

### 3.6. X128–133del Downregulates the Transcriptional Activity of CP/ENII through Attenuating the Binding Capacity of HNF4α to HBV CP/ENII

The corresponding nucleotide deletion of x128–133del is nt1755–1772del. As shown in Figure 5A, HBV nt1755–1772 is located at the overlapping region of the X region and CP/ENII region. It has been reported that there are two binding motifs of HNF4α in the HBV CP/ENII region, one is located at nt1660–1672, and the other is located at nt1755–1774 [31]. Therefore, the nt1755–1772del can lead to the deletion of the second binding motif of HNF4α in the CP/ENII region. To determine the effect of nt1755–1772del on CP/ENII transcriptional activity, the nt1755–1772del mutant reporter plasmid (pGL3-CP/ENII-Del) was first generated from the wild-type CP/ENII luciferase reporter plasmid (pGL3-CP/ENII). Next, pGL3-CP/ENII or pGL3-CP/ENII-Del plasmid, HNF4α expression plasmid (pCDH-flag-HNF4α) or vector control plasmid (pCDH), and renilla luciferase expression plasmid (pRL-TK), which was used as an internal control plasmid, were co-transfected into the Huh7 and HepG2 cells, and then the dual-luciferase reporter assays were performed to explore the effect of nt1755–1772del on the transcriptional activity of CP/ENII. As shown in Figure 5B, nt1755–1772del could significantly reduce the transcriptional activity of HBV CP/ENII, and the ability of ectopic HNF4α to enhance the transcriptional activity of the nt1755–1772del mutant CP/ENII was weaker than that of the wild-type CP/ENII, even though HNF4α could significantly enhance the transcriptional activity of either the wild-type or nt1755–1772del mutant CP/ENII. This indicates that nt1755–1772del could reduce the ability of HNF4α to enhance the transcriptional activity of CP/ENII.

Furthermore, ChIP-PCR confirmed that nt1755–1772del could attenuate the binding capacity of HNF4α to HBV CP/ENII (Figure 5C and Appendix A), which was further confirmed by ChIP-qPCR (Figure 5D). Therefore, nt1755–1772del downregulated the transcriptional activity of HBV CP/ENII through attenuating the binding of HNF4α to HBV CP/ENII.

## 4. Discussion

At present, HBV MTCT remains one of the predominant routes of new HBV infections worldwide, and the virological factors affecting HBV MTCT have not been fully clarified. Several studies have reported that the mutations in the X region of the HBV genome may play an important role in affecting HBV MTCT [12,22,23]. In this study, many amino acid mutations with high mutation rates (>10%) were found in the X region of the HBV genome, and obvious differences in the mutation rates of several amino acid mutations were found between the immunoprophylaxis failure and success groups.

It is well-known that HBx, as a transactivator relying on its C-terminal domain (aa 51–154), can promote HBV replication through directly promoting HBV transcription and indirectly regulating the expression of cellular transcription-regulatory factors [25,26,27,28]. HBx can also promote HBV replication through degrading host restriction factors such as Smc5/6 and regulating the epigenetic signatures of HBV cccDNA minichromosomes such as HBV DNA methylation and histone modifications of the cccDNA minichromosomes [32,33]. Moreover, the HBx C-terminal domain has been reported to be related to the stability of HBx protein [30], whereas the HBx N-terminal domain (aa 1–50) can inhibit the transactivation activity of the HBx protein [24,34]. The A1762T/G1764A double mutation is a hot-spot mutation in the overlapping region of the X region and CP/ENII region, which creates a binding site for the transcription factor hepatocyte nuclear factor 1 (HNF1) and two amino acid changes (K130M and V131I). The former change can result in the reduction in the HBeAg level by suppressing preC mRNA expression [18], and the latter change can enhance the activity of HBx to regulate hypoxia-inducible factor 1α (HIF-1α) and nuclear factor-kappa B (NF-κB), which might be related to the occurrence of hepatocellular carcinoma (HCC) [35]. Several studies have reported that the A1762T/G1764A double mutation can reduce the risk of HBV MTCT [17,18,19,20,21]. However, in this study, there was no significant difference in the incidence of the A1762T/G1764A double mutation between the immunoprophylaxis failure and success groups (13.64% vs. 6.67%, *p* = 0.490), which might be due to the fact that all mothers enrolled in this study were HBeAg positive. Therefore, the A1762T/G1764A double mutation might not be related to HBV MTCT in HBeAg-positive mothers.

Next, we found that HBx mutations (xG22S, xA23T, xG32R, and xS42Y) detected in the mothers of the immunoprophylaxis failure group were mainly located in the N-terminal domain, whereas the HBx mutations (xH86R, xM103L, and x128-133del) detected in the mothers of the immunoprophylaxis success group were mainly located in the C-terminal domain. Accordingly, the mutations of the HBx N-terminal domain might enhance the transactivation activity of HBx by weakening its inhibiting effect, which could promote HBV replication and contribute to immunoprophylaxis failure, while the mutations of the HBx C-terminal domain might inhibit the transactivation activity of HBx or affect its stability, which could inhibit HBV replication and contribute to immunoprophylaxis success. Further analyses showed that only the mutation rate of x128–133del in the immunoprophylaxis success group was significantly higher than that in the immunoprophylaxis failure group. As expected, x128–133del significantly weakened HBV replication, which could be partially rescued by wild-type HBx protein. These results indicate that x128–133del was not conducive to the establishment of new infection for HBV, and might contribute to immunoprophylaxis success. For the mechanism, x128–133del could significantly reduce the level of the HBx protein at the post-transcriptional level. Furthermore, we found that the level of the x128–133del mutant HBx protein could be increased to a level close to the wild-type HBx protein under the treatment of proteasome inhibitor MG132, but not for lysosome inhibitor HCQ, indicating that x128–133del reduced the HBx level by promoting the proteasome-dependent degradation of the HBx protein.

Meanwhile, the corresponding nucleotide mutation of x128–133del was nt1755–1772del, which could also lead to the deletion mutation in the HBV CP/ENII region because nt1755–1772 was located at the overlapping region of the HBV X and CP/ENII regions. It is well-known that HNF4α is an important transcription factor binding to the HBV CP/ENII region and regulating its transcriptional activity [36]. HBV nt1755–1772del could lead to the deletion of the second binding motif of HNF4α in the HBV CP/ENII region, and thus significantly reduce the transcriptional activity of CP/ENII. Although HNF4α could enhance the transcriptional activity of either the wild-type or nt1755–1772del mutant CP/ENII, the ability of HNF4α to enhance the transcriptional activity of the nt1755–1772del mutant CP/ENII was weaker than that of the wild-type CP/ENII. Furthermore, the binding capacity between the HNF4α and nt1755–1772del mutant HBV CP/ENII was confirmed to be attenuated, which should be due to the fact that nt1755–1772del made one of two HNF4α binding motifs disappear. Since the other binding motif of HNF4α was still present, nt1755–1772del could attenuate but not eliminate the enhancement effect of HNF4α on the transcriptional activity of CP/ENII by attenuating the binding of HNF4α to CP/ENII.

In conclusion, this study first found that x128–133del might improve the immunoprophylaxis effect of HBV MTCT through weakening HBV replication. In terms of mechanisms, the x128–133del (nt1755–1772del) mutation reduces the level of the HBx protein by promoting the proteosome-dependent degradation of the HBx protein, and reduces the transcriptional activity of CP/ENII by attenuating the binding capacity of HNF4α to CP/ENII. However, in this study, the potential effect of the HBV X region mutation on HBV MTCT was only explored in vitro due to the lack of animal models for HBV MTCT. Taken together, this study might be helpful in clarifying the virological factors affecting the immunoprophylaxis effect of HBV MTCT and formulating the optimal immunization strategy for children born to HBeAg-positive mothers.

## Figures and Tables

**Figure 1 viruses-14-01887-f001:**
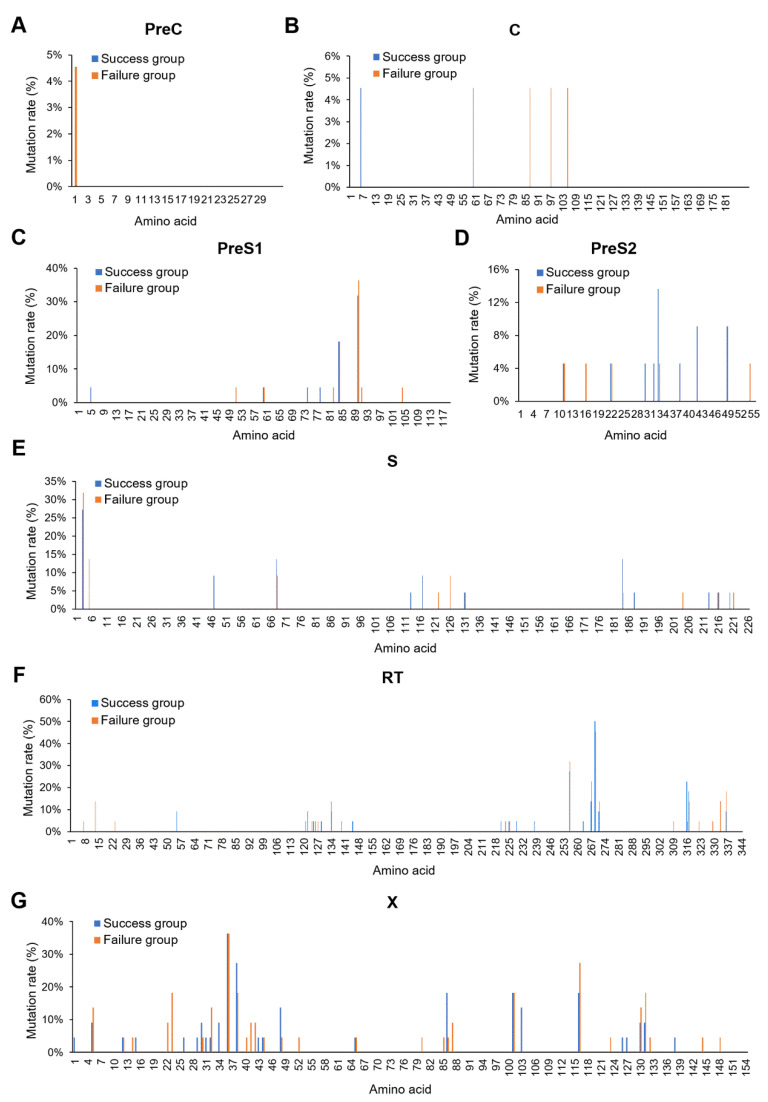
Amino acid mutations in the HBV coding regions. The amino acid mutation rates of the HBV preC (**A**), C (**B**), preS1 (**C**), preS2 (**D**), S (**E**), RT (**F**), and X (**G**) regions were analyzed in the mothers of the immunoprophylaxis failure and success groups. All of the mutations were defined on the basis of the same consensus sequence synthesized by all clones from the mothers.

**Figure 2 viruses-14-01887-f002:**
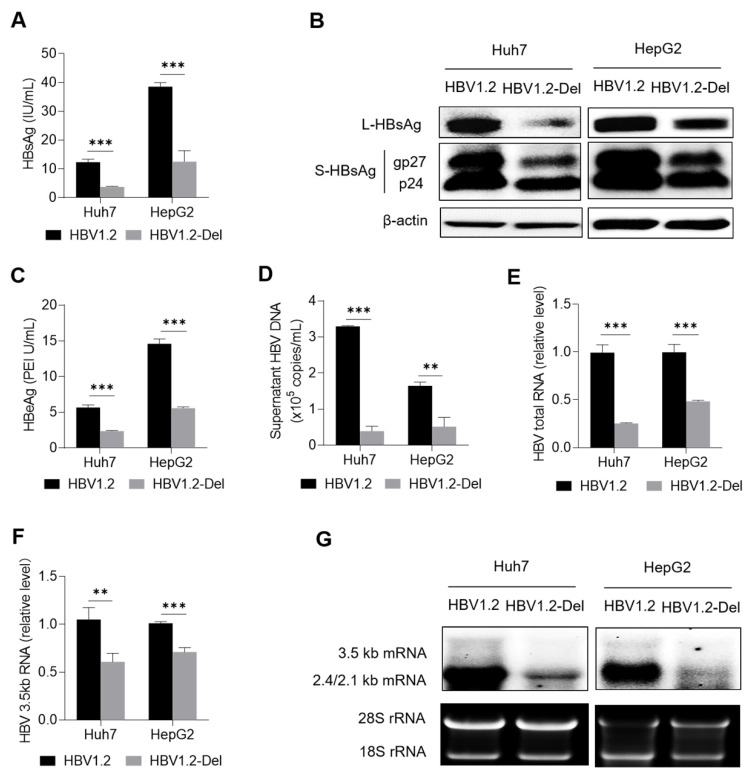
The effect of x128–133del on HBV replication. The pBB4.5-HBV1.2 or pBB4.5-HBV1.2-Del plasmid and pCDH-Nluc plasmid were co-transfected into the Huh7 and HepG2 cells (5 × 10^5^ cells/well in a 6-well plate). Cell culture supernatants and cells were harvested at 72 h after transfection. (**A**) The level of HBsAg in the cell culture supernatants was detected by the chemiluminescence immunoassay. (**B**) The intracellular levels of large HBsAg (L-HBsAg) and small HBsAg (S-HBsAg) including gp27 and p24 proteins were detected by Western blot. The β-actin protein was used as the internal control. (**C**) The level of HBeAg in the cell culture supernatants was detected by the chemiluminescence immunoassay. (**D**) The level of supernatant HBV DNA was detected by qPCR. The levels of intracellular HBV total RNA (**E**) and 3.5 kb RNA (**F**) were detected by RT-qPCR. The *ACTB* mRNA was used as the internal control. (**G**) The levels of intracellular HBV mRNAs including HBV 3.5 kb, 2.4 kb, and 2.1 kb RNAs were detected by Northern blot. The 28 s and 18 s rRNAs were used as the internal control. ** *p* < 0.01, *** *p* < 0.001, Student’s *t*-test.

**Figure 3 viruses-14-01887-f003:**
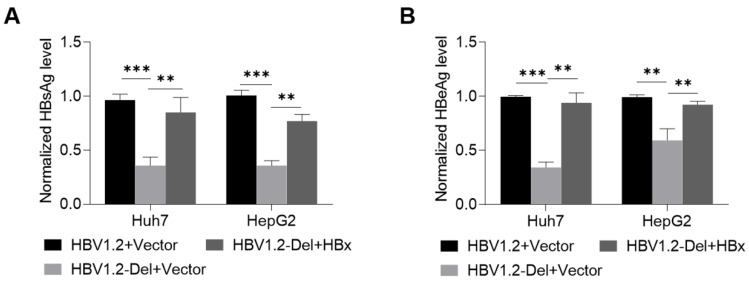
The effect of wild-type HBx protein on the x128–133del-weakened HBV replication. The pBB4.5-HBV1.2 or pBB4.5-HBV1.2-Del plasmid, pCDH-HBx-3×flag or pCDH vector control plasmid, and pCDH-Nluc plasmid were co-transfected into the Huh7 and HepG2 cells (5 × 10^5^ cells/well in 6-well plate), and then the levels of HBsAg (**A**) and HBeAg (**B**) in the culture supernatants of the Huh7 and HepG2 cells were detected by the chemiluminescence immunoassay at 72 h after co-transfection. ** *p* < 0.01, *** *p* < 0.001, Student’s *t*-test.

**Figure 4 viruses-14-01887-f004:**
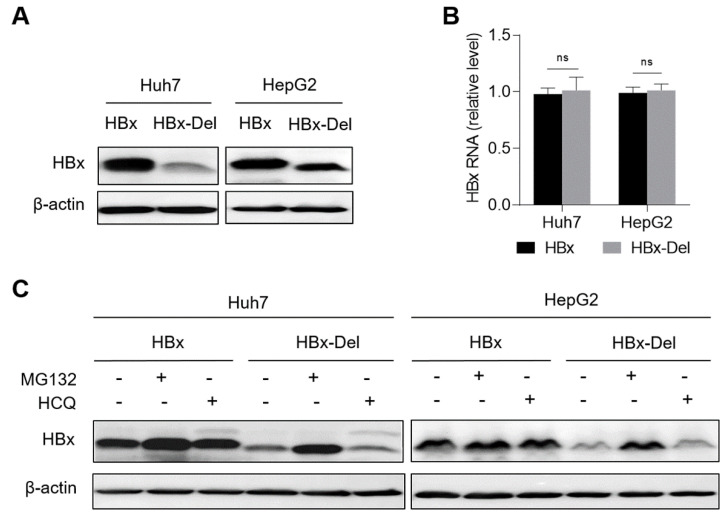
The effect of x128–133del on the HBx protein and mRNA levels. The pCDH-HBx-Del-3×flag or pCDH-HBx-3×flag plasmid was transfected into the Huh7 and HepG2 cells (5 × 10^5^ cells/well in 6-well plate), and then the cells were harvested at 48 h after transfection. (**A**) The level of HBx protein was detected by Western blot. The β-actin protein was used as the internal control, and (**B**) the level of HBx mRNA was detected by RT-qPCR, and *ACTB* mRNA was used as the internal control. (**C**) The level of HBx protein was detected by Western blot in the Huh7 and HepG2 cells transfected with pCDH-HBx-3×flag or pCDH-HBx-Del-3×flag plasmid and treated with 15 μM proteasome inhibitor MG132 for 6 h or 50 μM lysosome inhibitor HCQ for 12 h at 48 h after transfection. The β-actin protein was used as the internal control. HBx-Del—x128–133del mutant HBx protein; ns—no statistical significance, Student’s *t*-test.

**Figure 5 viruses-14-01887-f005:**
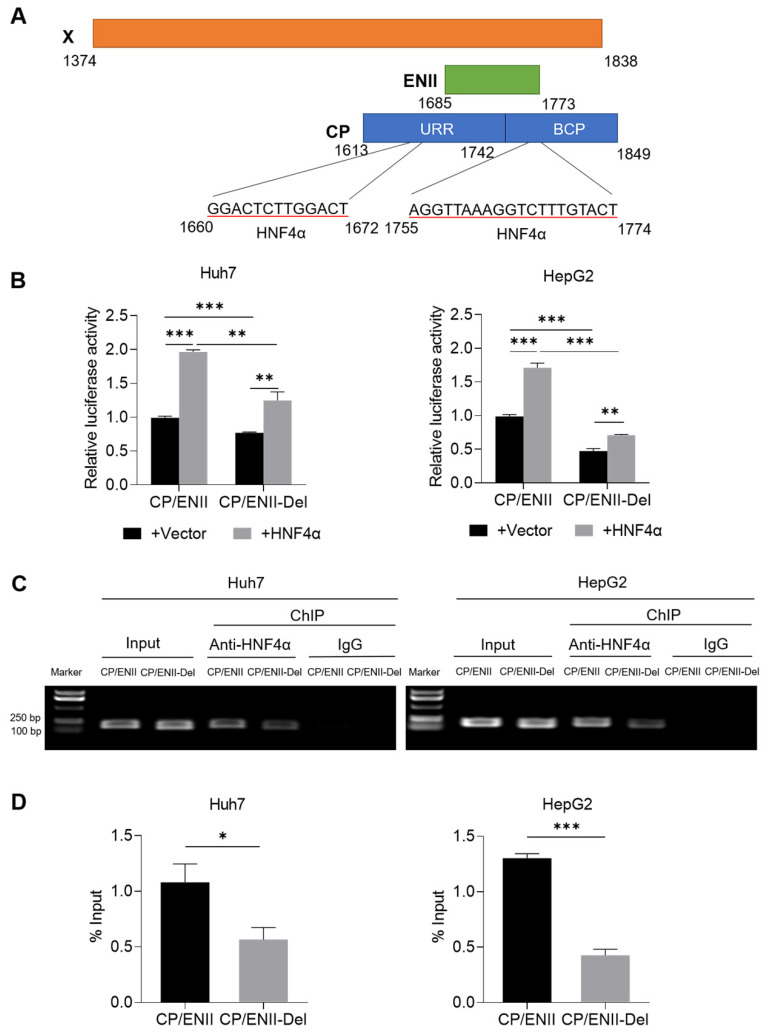
The effect of the nt1755–1772del on HBV CP/ENII transcriptional activity. (**A**) A diagram of the HNF4α binding sites in the HBV CP/ENII region. (**B**) The pGL3-CP/ENII or pGL3-CP/ENII-Del plasmid, pCDH-flag-HNF4α or pCDH vector control plasmid, and pRL-TK plasmid were co-transfected into the Huh7 and HepG2 cells (1 × 10^5^ cells/well in 12-well plate), and then the transcriptional activity of HBV CP/ENII was detected by dual-luciferase reporter assays at 48 h after co-transfection. The pCDH-flag-HNF4α and pGL3-CP/ENII or pGL3-CP/ENII-Del plasmids were co-transfected into the Huh7 and HepG2 cells (3 × 10^6^ cells/well in 10 cm dish), and then the binding capacity between HNF4α and HBV CP/ENII was detected by ChIP-PCR (**C**) and ChIP-qPCR (**D**) at 48 h after co-transfection. DNA ladder: DL5000. * *p* < 0.05, ** *p* < 0.01, *** *p* < 0.001, Student’s *t*-test. ENII—enhancer II; CP—core promoter; CP/ENII-Del—nt1755–1772del mutant CP/ENII; URR—upper regulatory region; BCP—basic core promoter; HNF4α—hepatocyte nuclear factor 4α; ChIP—chromatin immunoprecipitation.

**Table 1 viruses-14-01887-t001:** The baseline characteristics between the immunoprophylaxis failure and success group.

		Immunoprophylaxis Success Group	Immunoprophylaxis Failure Group	*p*
**Mothers**				
Number		22	22	
Age (years), median (range)		25.00 (20.00–35.00)	25.50 (19.00–34.00)	0.906
HBsAg (log_10_ IU/mL), median (range)		4.51 (3.78–4.91)	4.49 (3.49–4.88)	0.925
HBeAg (log_10_ S/CO), median (range)		3.15 (3.03–3.28)	3.16 (2.90–3.23)	0.907
HBV DNA (log_10_ IU/mL), median (range)		8.28 (7.18–8.72)	8.23 (7.48–8.96)	0.823
ALT (<40 U/L), n (%)		22 (100)	22 (100)	–
Genotype C2, n (%)		22 (100)	22 (100)	–
**Infants**				
Gender, male: female		9:13	13:9	0.228
Birth weight (kg), median (range)		3.30 (2.60–4.25)	3.55 (2.60–4.00)	0.204
Parturition manner, cesarean: vaginal		11:11	12:10	0.763
Feeding pattern, breast ^a^: artificial		8:14	7:15	0.750
Infant’s age at first dose of HepB (hours), n (%)	<12 h	22 (100)	19 (86.36)	0.232
	12–24 h	0	3 (13.64)	

^a^ Breast-feeding included mixed feeding.

**Table 2 viruses-14-01887-t002:** The mutations of the HBV X region in the immunoprophylaxis failure and success groups.

Nucleotide Mutation	Region	Amino Acid Mutation	Success Group, n (%)	Failure Group, n (%)	*p*	*Q* ^a^
G1437A	X	xG22S	0	2 (9.09%)	0.023	0.052
G1440A	X	xA23T	4 (3.33%)	4 (18.18%)	0.023	0.052
T1464C	X	xS31P	7 (5.83%)	0	0.531	0.534
G1467C	X	xG32R	3 (2.50%)	3 (13.64%)	0.048	0.086
C1498A/A1499C	X	xS42Y	0	2 (9.09%)	0.023	0.052
G1515A	X	xD48N	15 (12.50%)	1 (4.55%)	0.473	0.534
A1630G	X/CP	xH86R	14 (11.67%)	1 (4.55%)	0.534	0.534
A1680C	X/CP	xM103L	7 (5.83%)	0	0.531	0.534
1755–1772del	X/CP/ENII	x128–133del	33 (27.50%)	0	0.005	0.045

^a^ Correction for multiple comparison was analyzed by the Benjamini–Hochberg test.

## Data Availability

Data not published in the manuscript are available from the corresponding author.

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
