# Peer review of "HBx 128–133 Deletion Affecting HBV Mother-to-Child Transmission Weakens HBV Replication via Reducing HBx Level and CP/ENII Transcriptional Activity"

_viruses, 2022, doi:10.3390/v14091887_

Round 1
Reviewer 1 Report (Previous Reviewer 2)
Sorry if I'm late, but these days in my country is mid-August traditional feast.
Despite the fact that the comments and suggestions are from another reviewer, I believe that the manuscript has been significantly improved and now deserves publication in "Viruses". I still have some minor concerns about English, e.g. the verb OCCURS (third line of the Abstract) does not seem appropriate to me, although the meaning is clear. Maybe, if the sentence began with "In some infants born.....still occur...and they develop... Or find another verb instead of occur. But the Editor will be able to do the best thing.
Reviewer 2 Report (Previous Reviewer 1)
Dr. Song and colleagues have appropriately revised their manuscript and successfully integrated my comments. I have no further comments.
This manuscript is a resubmission of an earlier submission. The following is a list of the peer review reports and author responses from that submission.
Round 1
Reviewer 1 Report
Dr. Song and colleagues presented an interesting retrospective and experimental study on the analysis of HBx mutations potentially associated with immunoprophylaxis and affecting HBV transmission from mother to child in HBsAg-positive individuals from two provinces in China. The colleagues examined 1694 HBsAg-positive pregnant without antiviral treatment. Their infants received prophylactic doses of HepB and HBIG, and 29 infants were still HBsAg positive at 7 months of age indicating an immunoprophylaxis failure. For further analyses 22 infants and their mothers with immunoprophylaxis failure were compared and matched with 22 infant/mother pairs with immunoprophylaxis success. The isolated HBV genomes were analysed by direct sequencing. Mutations of interest of the HBV genomes were introduced into HBV replicons and further evaluated for HBV replication competence in cell culture experiments. As a result, the authors demonstrated that a deletion in the HBx domain (x128-133del) might be related to the effects of immunoprophylaxis and thus could be responsible for HBV MTCT by attenuating HBV replication.
Overall, the study is well conducted, written mostly concisely, and is a good addition to the knowledge of HBV-MTCT. The cell culture experiments are well designed and the results are convincing. However, there are some points that should be addressed.
Specific comments
1. Some sentences are difficult to understand, e.g. in the discussion section "Summary" first sentence (lines 589 ff). Therefore, the English should be checked by a native English speaker to read it better.
2. The two groups immunoprophylaxis failure and success should be explained and defined more precisely.
3. The HBV genotypes of the mothers studied were C2. What about the remaining pregnant women. Is the outcome possibly an effect of HBV genotype C2.
4. The authors sequenced the whole HBV genome and focused on the HBx and CP domains. What about the preS/S region? Were there any relevant mutations in the region that should be noted?
Reviewer 2 Report
Elegant research, clear exposition, although for the moment I do not see a practical application, this does not detract from its importance.
Author Response
Elegant research, clear exposition, although for the moment I do not see a practical application, this does not detract from its importance.
Response: Thanks a lot for the reviewer’s encouragement of our work.

Reviewer 3 Report
In this article, Song et al reported the association between HBx 128-133 deletion and reduced risk of mother-to-child transmission of hepatitis B virus. Moreover, it was found that this deletion reduced the level of HBx protein and down-regulated the transcriptional activity of core promoter/enhance II, which resulted in milder viral replication.
As mother-to-child-transmission is an important route of HBV infection which mostly resulted in chronic infection, the value of this study is evident. However, extra-care should be taken to draw a conclusion on the effect of viral sequence variation on the rate of MTCT. As shown in Table2, a total of 10 variants were analyzed for its association with MTCT, correction for multiple comparison should be used to reduce the possibility of false discovery. Indeed, the effect of HBX 128-133 would be much clearer if the author could reanalyze the sequence in the complete cohort of immnoprophylaxis success group (over 1000 cases) using targeted PCR sanger sequencing and compare its frequency with the 22 cases of immunoprophylaxis failure group.
Regarding the effect of HBx 128-133 deletion on viral replication, the authors mainly used co-transfection assays. Additional internal control should be used to make sure the transfection efficiency between groups are comparable. One can use a reporter construct expressing secreted luciferase to co-transfect with HBV plasmid, which can indicate the efficiency of plasmid delivery into cells.
Finally, in Figure 5C-D, quantitative PCR should be used to compare the immunoprecipitated minichromosome, the use of endpoint agarose gel electrophoresis may cause bias.
Minor issue:
1. Figure 4, The author used FLAG tagged HBx expression construct but in the western blot, anti-HBx antibody was used. Can the author clarify which antibody was used?
2. Additional language editing is needed for the text.
Round 2
Reviewer 3 Report
It seems that the authors are unwilling to consolidate their conclusion by doing additional sequencing on additional samples in the immunoprophylaxis success group. I strongly disagree with the authors in point 1, corrections must be made when doing multiple comparisons. Regarding the internal controls in the transfection assays, I did not only mean the luciferase reporter assay but all the co-transfection assays including those shown in Fig2 and 3.